# Specific Resonant Properties of Non-Symmetrical Microwave Antennas

**DOI:** 10.3390/s21030939

**Published:** 2021-01-31

**Authors:** Ján Labun, Pavol Kurdel, Alexey Nekrasov, Mária Gamcová, Marek Češkovič, Colin Fidge

**Affiliations:** 1Faculty of Electrical Engineering and Informatics, Technical University of Košice, Letná 9, 04200 Košice, Slovakia; jan.labun@tuke.sk (J.L.); maria.gamcova@tuke.sk (M.G.); 2Faculty of Aeronautics, Technical University of Košice, Rampová 7, 04121 Košice, Slovakia; marek.ceskovic@tuke.sk; 3Institute for Computer Technologies and Information Security, Southern Federal University, Taganrog 347922, Russia; alexei-nekrassov@mail.ru; 4Science and Engineering Faculty, Queensland University of Technology (QUT), Brisbane, QLD 4001, Australia; c.fidge@qut.edu.au

**Keywords:** microwave antenna, antenna impedance, antenna resonance

## Abstract

The aircraft avionics modernization process often requires optimization of the aircraft itself. Scale models of aircraft and their antennas are frequently used to solve this problem. Here we present interesting properties of the resonant antennas, which were discovered serendipitously during the measurement process of some microwave antennas’ models as part of an aircraft modernization project. Aircraft microwave antennas are often designed as non-symmetric flat microwave antennas. Due to their thin, low and longitudinally elongated outer profile, they are also called tail antennas. An analysis of the resonant properties of non-symmetric antennas was performed in the band from 1 GHz to 4 GHz. The length of the antenna models ranged from 2 cm to 7 cm. The width of the antennas, together with the thickness of the strip, was always a constant parameter for one measured set of six antennas. In the measurement and subsequent analysis, attention was focused on the first-series resonant frequency (*λ*/4) of each antenna. During the evaluation of the resonance parameters, the flat microwave antenna models showed specific resonant properties different from those of conventional cylindrical microwave antennas. This article aims to inform professionals about these unknown specific properties of non-symmetrical antennas. The results of experimental measurements are analyzed theoretically and then visually compared using graphs so that the reader can more easily understand the properties observed. These surprising observations open up some new possibilities for the design, implementation, and use of flat microwave antennas, as found in modern aircraft, automobiles, etc.

## 1. Introduction

In the process of modernizing new aircraft avionics systems, it is often necessary to install another antenna on the aircraft’s fuselage. Determining an optimal position for the new aircraft antenna is a key problem. At present, the solution to this problem usually starts with a computer simulation. After the introductory suggestion of an optimal position, experimental measurements on an aircraft scale model, with the use of a miniaturized antenna model, are performed. To perform such measurements, the aircraft model as well as the antenna model must be miniaturized. In this way, it is necessary to transform the theory and practice of antennas from a full-size aeronautical frequency band in MHz to a frequency range in the microwave band [1,2]. The miniaturized antenna model working in a resonance frequency in the GHz range must possess certain parameters (characteristics, VSWR, etc.) identical with the real aircraft antenna working in the MHz range [3].

This paper deals with interesting resonance characteristics of such antennas, which were discovered serendipitously during measurements on models of small microwave antennas in the GHz range performed as part of the process described above. Analysis of the resonance characteristics of non-symmetrical microwave antennas was performed in frequency ranges from 1 GHz to 4 GHz. The common parameter of the antenna set measured was their transverse dimension, shape, and material [4,5,6]. The length of the antennas, which was the variable parameter, ranged from 2 cm to 7 cm. During the analysis, the focus was on the first series quarter-wave resonance of each antenna. When doing summary analysis in the common graph of measured parameters, the microwave antennas revealed resonance characteristics that are not shown by classical cylindrical antennas. As simulation programs for antenna characteristics are based on the theory of mainly classical antennas, the new knowledge presented in this paper provides an important basis for updating such simulation programs. Ultimately this information, derived from small-scale MHz range observations, applies to full-size flat microwave antennas for automobile systems or UAVs in higher frequencies in the GHz range.

The paper is structured as follows. The first three short sections present the purpose of the experimental measurements during which the specific resonance characteristics of flat microwave antennas were observed. In the introductory subsection of the fifth section of this paper a brief theory of microwave antennas focused on the problem solved in the paper is presented. Next, the second sub-section confirms this well-known theory with the results measured on classical cylindrical antennas. The final, third sub-section reveals the new, unknown resonance characteristics we discovered from the results of the measurement of flat microwave antennas. The presented findings open certain new possibilities to design small microwave antennas for automotive and aviation technology in the higher GHz frequencies [7,8,9].

## 2. Materials and Method

Here we describe the materials that were prepared for measuring resonant properties of microwave antennas and our method of processing the results, based on well-known antenna theory.

### 2.1. Material for Measuring the Properties of Microwave Antennas

If the antenna model is to radiate around the model aircraft in the same way as the corresponding antenna on an actual aircraft, both antennas must have approximately the same shape, proportional dimensions, similar electrical properties, etc. However, implementing such an antenna model would be complicated and time-consuming. For a single use, it would be also too costly. To make this problem more efficient and easier to solve, the research team instead created several sets of antenna models, with different shapes, different lengths, and different transverse dimensions [10]. An illustration of three-part sets of two antenna models (cylindrical and flat types) is shown in Figure 1a. Twelve six-piece sets of the produced antenna models are shown in Figure 1b. For the purposes of this article, three additional sets of flat microwave antenna models were produced from 0.35 mm thick brass sheet. For good handling during the measurements, thin antennas were prepared for their quick and easy clamping, and a conductive artificial ground measuring 10 × 10 cm was used [11].

During the measurements, the VSWR value of the individual antennas was evaluated in the frequency range from 1 GHz to 4 GHz. The minimum value of VSWR on the frequency axis determined the required resonant frequency of the antenna model. The VSWR record of the antenna for determining the value of the sought resonant frequency is shown in Figure 2, where the first (quarter-wave) resonant frequency is 2063.29 MHz (point M1) and the second (half-wave-parallel) resonant frequency is 3645.57 MHz (point M4) [12,13,14,15].

After each evaluation of the resonant frequencies of one set of six antennas, the results were plotted as shown in Figure 3. The sequence number of Record 1 corresponds to the longest antenna (low frequency) and the sequence number of Record 6 corresponds to the smallest antenna (high frequency) [16].

### 2.2. Method for Evaluating Resonant Properties of Microwave Antennas

From a practical point of view, this article focuses on the frequency dependence of an antenna to a change in its mechanical dimensions. Specifically, it is a matter of determining the resonant frequencies of a set of antennas of certain lengths. Each antenna is in resonance under certain conditions related to its impedance. To explain the innovative view of this article towards this problem, it is necessary to outline a brief theory of the impedance of an antenna $Z_{A}$. Let’s consider the impedance of the antenna at its input terminals $Z_{IN}$. The input impedance of the antenna generally includes the radiation impedance $Z_{\Sigma }$ and the loss impedance $Z_{LOS}$:(1)$$Z_{A}=Z_{IN}=Z_{\Sigma }+Z_{LOS}$$

At frequencies in the UHF band, the antennas are realized as self-supporting and are made of a well-conducting material. Therefore, the losses in this band are negligible, so we can determine $Z_{LOS}=0$. In general, the impedance is a complex number that has a real *R* and an imaginary *X* component. Then, the radiation impedance is defined as:(2)$$Z_{\Sigma }=R_{\Sigma \;}\pm jX_{\Sigma }$$
where *R*_Σ_ is the radiation resistance and *X*_Σ_ is the reactance, which is the reactive component of the impedance. The radiation resistance takes the power from the source, proportional to the power that the antennas emit. The radiation reactance takes power from the source that is not radiated in reality. The lines of force of the electric and magnetic field originate and disappear, but they do not detach from the antenna. From this point of view, reactance is an undesirable component. The specific value of the impedance depends on the wavelength ratio of the operating frequency *λ* and the length of the antenna arm l, so we speak of the l/λ ratio. In general, this impedance dependence has a periodic character, and its simplified course is shown in Figure 4. The radiation resistance (blue course) changes periodically with the l/λ ratio, but it always has a positive value. The reactance (red line) also changes periodically with the l/λ ratio, from positive to negative values. With positive values, the antenna has an inductive character (inductance $X_{L}$). At negative values, the antenna has a capacitive character (capacitance $X_{C}$) [17].

The resonance of the antenna occurs whenever the course of the reactance intersects the zero axis, $X_{\Sigma }=0$. In this situation, the radiation impedance is proportional only to the radiation resistance, $Z_{\Sigma }=R_{\Sigma }$, and the antenna emits HF energy with the minimum possible losses. This situation is periodically repeated for each whole multiple of the ratio l/λ = 0.25. According to the ordinal number of the multiple of the given ratio (1, 2, 3, 4, ...), we speak of the first, second, third, fourth, etc., antenna resonance. In antenna practice, only the first and second resonances are used, because unsuitable radiation characteristics of the antenna are formed at other resonances. A more detailed graphical representation of only these two resonances is shown in Figure 5a.

In Figure 5a, the highlighted course represents the dependence of the thickness of an antenna to its impedance. This course represents the regularity when a thinner dipole (dotted lines) has a more significant change of the value of impedance on the ratio of l/λ and intersects the zero-impedance axis with a steeper slope than the thicker dipole. At the same time, it can be observed in Figure 5 that these two dipoles do not intersect the horizontal axis in one place. The perpendicular vertical axes represent the position of the exact ratio of l/λ = 0.25; 0.5; 0.75, etc. However, the antenna resonance does not occur exactly at these ratios, but a little earlier. The exact position of this point depends on the slenderness of the antenna Ω**. The slimness of the antenna is expressed by the ratio of the length of the antenna arm *l* to its diameter *ϕ* or, in general, to the cross-sectional dimension of the antenna arm Ω=l/ϕ [18].

From a practical point of view, this means that the antenna is not in resonance when its length is exactly *λ/4*, but when it is slightly shorter. This shorter value is expressed by the antenna shortening factor *ξ* and it represents the ratio of the actual resonant length of the dipole (e.g., 0.22) to the ratio l/λ. For instance, when l/λ = 0.25 the factor would be *ξ* = 0.88. In practice, this value is in the range from 0.98 to 0.8, but in general aviation, for some type of antennas, it could be less. For aviation applications, in most cases, only the first resonance is used, because it has the smallest resonant length of one arm of the antenna dipole (unipole). In aviation, there is a natural requirement for minimum aerodynamic drag produced by an antenna mounted on the fuselage. Also, for these reasons, the first resonant frequency of each antenna was evaluated in the experimental measurements. Their resonance points on the axis of the ratio l/λ, from the reactant’s course of the impedance, were evaluated, as shown in Figure 5b. The radiation resistance value of the measured antenna was not evaluated for the purposes of this article. To theoretically clarify the specific properties of flat microwave antennas, as the ideological goal of this article, it is necessary to know the procedure for determining the slope of the reactance line and determine the position of resonance points R on the l/λ axis around the first resonance [19].

The known position of the two points was used to determine the slope of the reactance line. The first of these points is the generally known position of the value of the radiation impedance of the antenna at the exact length of one of its arms l=λ/4=0.25, i.e., [20]:(3)$$Z_{\Sigma }=73.1\;+\;j42.5\;\left[\Omega \right]$$

The position of this point is still the same even when changing the slimness coefficient of the antenna. We can therefore assume that the course of the reactance of each antenna measured by us passes through this point. The position of the second point in the graph is determined from the measured resonant frequency at which the antenna is in resonance and the subsequent calculation of the value of ratio l/λ. From the generally known antenna reactance waveforms (in the region of the 1st resonance), this waveform is almost linear in the range of the above points, as per Figure 5a,b. Since we have defined the position of two points for each measured antenna, we can thus display the lines passing through them, which are an important part of the course of their inductance in the region of the first-quarter wave resonance. Figure 6 shows unmeasured but indicative reactance lines of three antennas of different slimness. The positions, waveforms, and steepness of the individual reactance lines were determined by the technique described above. According to this developed theory, the following section shows the actual results of the reactance course of the measured antennas.

The position of the resonant points of the antennas can theoretically be found in Figure 6 in the range of the l/λ ratio from 0 to 0.25. But, practically, the resonance points can only be in two-fifths of this section, i.e., from 0.15 to 0.25. For clarity, in the following figures of a similar type in this section, the displayed area is that shown in the lower part of Figure 6 highlighted by a thick green line. The individual points of the resonant ratio l/λ on the horizontal axis and the slope of the line present interesting data. In aviation, from an aerodynamic point of view, it is necessary to implement an antenna with the shortest possible length. At the same time, this antenna must have the required range of operating frequencies at these minimum physical dimensions. With the above knowledge of antenna theory, we entered into the required experimental measurements with the goal defined in the introduction above. It is well known that thicker antenna wires of the same length as thinner antenna wires always have a lower resonant frequency. This rule applies when comparing them at any length. This fact is expressed by the illustrative graph in Figure 7. The parameter is the diameter of the antenna conductors, where it is planned to use diameter values of *ϕ* = (6, 12, 25) mm. It is clear from the sketched graph that on each equal length of the antennas, their resonant frequency can be defined by the relation: $f_{res}$
*ϕ*6 > $f_{res}$
*ϕ*12 > $f_{res}$
*ϕ*25. Such a simplified expression of resonant frequency ratios is taken as regularity in antenna theory. We also relied on this rule at the beginning of the production of our antenna models [18,21].

To apply the above theory, the graph in Figure 7 introduced antenna diameters through the antenna slimness coefficient Ω. The coefficient of the slimness of the antenna expresses the ratio of the length of the antenna *l* to its diameter ϕ, (Ω = *l*/ϕ), which means that the larger the value of Ω, the slimmer the antenna is. In the case of our measurements, we consider antenna wires with a diameter of *ϕ* = (6, 12, 25) mm. The slimness coefficient is different for each length. However, the property applies to the listed cylindrical antennas ($\Omega_{\phi 6}$ > $\Omega_{\phi 12}$ > $\Omega_{\phi 25}$).

## 3. Results and Discussion

In this section, we firstly present the well-known results of measuring resonant properties of classical cylindrical antennas. Then we present our new findings based on experimental measurements of the resonant properties of flat microwave antennas.

### 3.1. Resonance Properties of Measured Classical Cylindrical Antennas

Our original research aimed to experimentally verify the above facts, i.e., how the dimensions, shape and material of their components affect the resonant frequencies of antennas. Results similar to the illustrated graphs in Figure 7 were expected, but with real measured values of resonant frequencies, as per Figure 8. In these graphs it is possible to quickly orientate for the purpose of the production of antenna models, operating in the required frequency band, to use them in measurements on aircraft models. Although the implemented and measured sets included antennas with six different diameters *ϕ* = (6, 8, 10, 12, 15, 25) mm, only antennas with three diameters *ϕ* = (6, 12, 25) mm are used for the graph plots below in this article.

Based on the measured values of the resonant frequency of each antenna, the corresponding wavelength, $\lambda_{res}=c/f_{res}$, was determined from a known relationship. By defining the wavelength of the resonant frequency $\lambda_{res}$ for a particular antenna in terms of its length *l* and diameter *ϕ*, it was possible to determine its ratio$\;l/\lambda_{res}$. Since this was the first resonant frequency (also called the serial or quarter-wave frequency), the value of this ratio is always less than 0.25. The calculated values of resonance points as values of the ratio $l/\lambda_{res}$ of individual types of measured cylindrical antennas are shown in Table 1.

Based on the explanation of the general impedance dependence of the antenna on the l/λ ratio, presented in Figure 5 and Figure 6, only three calculated values of $l/\lambda_{res}$ from the measured values in Figure 6 were selected from Table 1 and subsequently plotted in the graphs in Figure 7. To be able to align the measured values of the antennas with the respective values given in Table 1 with their graphical representation in Figure 8 and Figure 9, the numerical values and their respective graphic waveforms are highlighted and aligned. Figure 9 shows the resonant points of cylindrical antennas on the axis of the $l/\lambda_{res}$ ratio, when the respective antenna is in resonance and thus when the reactance value of the antenna is *X* = 0. The resonant points corresponding to a thin antenna with a conductor diameter *ϕ* = 6 mm are presented in Figure 9a. The resonant points corresponding to an antenna conductor with a mean value of the diameter of *ϕ* = 12 mm are shown in Figure 9b. The resonant points corresponding to a thick antenna with a conductor diameter *ϕ* = 25 mm are in Figure 9c. Depending on the length of the antenna (shown only for 20, 50 and 70 mm), each trio of antennas of the respective diameter (6, 12, 25) mm forms a slightly slim, slim, and very slim cylindrical antenna:

Low-height antennas (*l* = 20 mm) are slightly slim antennas Ω***_ϕ_**_6_*
*l_20_, Ω**_ϕ_**_12_*
*l_20_, Ω**_ϕ_**_25_**l_20_*.

Medium height antennas (*l* = 50 mm) are slim antennas Ω***_ϕ_**_6_*
*l_50_, Ω**_ϕ_**_12_*
*l_50_, Ω**_ϕ_**_25_**l_50_*.

High-height antennas (*l* = 70 mm) are very slim antennas Ω***_ϕ_**_6_*
*l_70_, Ω**_ϕ_**_12_*
*l_70_, Ω**_ϕ_**_25_**l_70_*.

From the analysis of Figure 9, we can see that by reducing the length of the antenna its resonant ratio $l/\lambda_{res}$ (resonant frequency) is shifted to a lower value. By subsequently increasing the cross-section of the small antennas (20 mm), their resonant ratio $l/\lambda_{res}$ (resonant frequency) shifts to an even lower value. In the case of the whole measured set of 18 cylindrical antennas, the position of the resonant points ranges from a maximum of 0.245 to a minimum of 0.181, which represents the total extent of their change of 0.064 (dimensionless number). This fact is generally known in the theory of antennas and is used in antenna technology in the design of antennas. The effect of the change in the slimness of the cylindrical antenna (in the form of a change in the diameter of the conductor) with a constant height on the position of the resonant point can be generalized from Figure 9. It is also possible to observe the effect of altering the slimness of the cylindrical antenna (by changing the height of the conductor) with a constant diameter of the position of the resonance point. This fact can be assessed in the following Table 2.

For the characteristic course of changing the position of the resonant points of the whole set of classical cylindrical antennas, it is possible to emphasize the problem of frequency band coverage. The whole set consists of 18 measured antennas. These are divided according to the diameter of the conductor into three sets, each of six antennas. In terms of the frequency range, we consider only the values of the resonant points for the two “edge” (coarsest and thinnest) antennas, in each set:(1)Set of antennas *ϕ* = 6 mm   Figure 9a 0.245–0.196 = 0.049(2)Set of antennas *ϕ* = 12 mm Figure 9b 0.233–0.189 = 0.044(3)Set of antennas *ϕ* = 25 mm Figure 9c 0.222–0.181 = 0.041

Although the values of the difference of the resonance point Δ ($l/\lambda_{res}$) of classical cylindrical antennas differ (0.049, 0.044, 0.041), the differences according to the conductor diameter are not very significant. The total difference in the values of the resonant points of the whole set of cylindrical antennas is relatively small and is 0.064. For simplicity of illustration and clarity of the explanation of the problem in Figure 9, every graph shows the changes in the position of the points of the resonant ratio $l/\lambda_{res}$, but only for three antennas. This simplification gives the impression that the resonant points of antennas is based on known knowledge of antenna theory only. However, with this form of imaging, but using a larger number of resonant points of the antennas, as is found with non-cylindrical antennas, the images become more complex, confusing, and harder to interpret.

Therefore, another non-traditional form of displaying a larger number of resonant points of antennas was chosen (Figure 10). In Figure 10, 18 resonant points are shown, distributed according to the value of the $l/\lambda_{res}$ ratio, taking into account the length and diameter of the antenna. This explains the essence of the issue addressed in this article. Diagrams like that in Figure 9 show a concentrated distribution of resonant points, based on existing knowledge of antenna theory. However, the form of distribution of resonant points used in Figure 10 will allow us to better understand the unknown specific properties (differences) between the theory of flat microwave antennas from the theory of classical cylindrical antennas.

Figure 10 represents a concentrated distribution of the position of the resonant points of the antennas when their slimness changes, in the form of a change of height or diameter. The display of the three antenna diameter values is on three parallel horizontal axes. The fourth lower horizontal axis is a measure of the l/λ ratio. The length of the antenna is shown in the figure as a parameter. Each of the three resonant points of the same length is connected by a dashed line.

From Figure 9, it can be observed that when changing the slimness of the antenna and increasing its diameter, the resonant point of the antenna shifts to the left towards lower values of the l/λ ratio. This shift is proportionally the same on all lengths and all diameters of antennas. The inclination of the individual dashed lines that this shift expresses is almost parallel. Small differences correspond to measurement inaccuracies, like incorrect reading of the resonant frequency, manufacturing inaccuracy of the antenna, changes in the position of the antenna mounting, etc.

### 3.2. Resonance Properties of Measured Flat Microwave Antennas

In our general determination of the resonant properties of antennas, measurements were performed on various types of antennas, like cylindrical, square, and rectangular. Measurements made on flat metal material of various thicknesses were also performed. These antennas are classified as flat microwave antennas. After manufacturing complete sets of these antennas with the appropriate dimensions, their resonant properties were measured and recorded. During the measurements (made by observing discrete values of resonant frequencies), it became apparent that the way frequency values change based on length in flat microwave antennas is different than antennas with cylindrical cross-sections. This can be demonstrated on a graph (Figure 11). Some types of flat microwave antennas, whose cross-sectional dimensions in both planes are not the same, show similar resonance dependences but are still different from cylindrical antennas.

When enlarging only one of the transverse dimensions of the flat microwave antenna (like width), the course of the change of the frequency dependence on the length of the antenna does not have the same parallel curvature as for cylindrical antennas (Figure 8). In the case of the flat microwave antennas and other flat antennas, each set of antennas has a different curvature, a different steepness and, consequently, they intersect at one point. For the purposes of this article, we call this crossover value a “significant” point and it is marked with the letter *S*. Based on the measured values of the resonant frequency of each antenna, the corresponding wavelength was determined similarly to cylindrical antennas. By knowing the wavelength of the resonant frequency and knowing the specific length of the antennas, it was possible to determine the $l/\lambda_{res}$ ratio, which determines its resonant point. The values of resonance points calculated in this way of individual types of measured flat microwave antennas are shown in Table 3.

Figure 12 shows the resonance points of flat microwave antennas when the reactance value of the antenna is zero. In Figure 12a are points corresponding to a thin antenna with a strip width w = 5 mm. The points corresponding to a medium-thick antenna with a bandwidth *w =* 10 mm are in Figure 12b. The points corresponding to a thick antenna with a bandwidth *w =* 30 mm are shown in Figure 12c. Depending on the length of the antenna, *l =* (20, 55, 70) mm, each triple of antennas of corresponding strip width *w =* (5, 10, 30) mm forms a low-slim, slim, and very slim flat microwave antenna. Even in this case, antennas with a small height (20 mm) are low-slim antennas with slimness coefficients Ω*_w5_ l_20_, Ω_w10_ l_20_, Ω_w30_ l_20_*. Medium height antennas (55 mm) are medium slim antennas with slimness coefficients Ω*_w5_ l_55_*, Ω*_w10_ l_55_*, Ω*_w30_ l_55_*. High height antennas (70 mm) are very slim antennas with slimness coefficients Ω*_w5_ l_70_*, Ω*_w10_ l_70_*, Ω*_w30_ l_70_*.

In the case of the whole measured set of 18 flat microwave antennas, the value of the resonance point changes from a maximum of 0.247 to a minimum of 0.139, which represents the total range of 0.108. This fact is known in antenna theory and used in antenna technology for the realization of small broadband antennas. The influence of the slimness coefficient change Ω** of the flat antenna on the change of the ratio *l**/**λ_res_* due to changing its width *w* at the same height *l*, or due to changing its height *l* at the same width *w* can be seen in Figure 12. Also, it can be quantified from Table 4.

For the characteristic course of the change of the position of the resonant points of the whole set of flat microwave antennas, it is possible to assess the problem of frequency band coverage. Also, in this case, the whole set is represented by our 18 measured antennas, which are divided according to the width of the strip into three sets of six antennas. In terms of the frequency range, we also consider here only the values of the resonant points for the two “edge” (widest and narrowest) antennas from each set.

(1)Set of antennas *w* = 5 mm   Figure 12a 0.231–0.204 = 0.027(2)Set of antennas *w* = 10 mm Figure 12b 0.24–0.171 = 0.069(3)Set of antennas *w* = 30 mm Figure 12c 0.247–0.139 = 0.108

The values of the difference of the resonant points Δ (l/λ_res_) of the flat microwave antennas are very different (0.027, 0.069, 0.108) and in this case, the differences according to the bandwidth are very significant. The difference in the values of the resonant points of the whole set of flat microwave antennas is determined by the antenna with the widest strip and it is relatively high at 0.108.

Many of these results, with respect to the operating frequency and bandwidth of conventional cylindrical antennas and flat microwave antennas, are relatively well known. However, what was previously unknown is that the graphs of the dependence of the resonant frequencies of the flat microwave antennas on their length intersect each other, like the case shown in Figure 11. At the same thickness of the conductive strip from which the antennas are made and when displaying their frequency dependence on their length (where the width of the strip is the parameter), their characteristics intersect at one significant point *S*. Conventional cylindrical antennas in the same conditions have no such point and display parallel characteristics like that in Figure 8.

Like in the previous case of the cylindrical antennas (Figure 10), in this case of displaying the results for flat microwave antennas (Figure 13) we also have chosen a non-traditional form of a graph for a large number of resonant points of our set of 18 flat microwave antennas. This gives us an original view of the frequency dependence of flat microwave antennas on their thickness, width and length as shown. Although developed earlier based on classical cylindrical antennas, this new view of flat microwave antennas properties is better for understanding the new observations on such antennas, as per Figure 10 and Figure 13.

By comparing Figure 8 with Figure 11 as well as with Figure 10 and Figure 13, it can be seen that flat microwave antennas do not behave in the way predicted by classical antenna theory. General antenna theory was developed based on the measurement of cylindrical antennas, not on the variety of shapes encountered in modern aircraft and used in our experiments. Crossed graphical waveforms (resonant frequency dependence on antenna length, where the parameter is the width of the strip) has created a significant point *S* in Figure 11. This point *S* can be also observed in Figure 13 but in a slightly different position. In the case of cylindrical antennas, as the conductor diameter increases, all the original resonant points move proportionally to the lower values of the *l*/*λ* ratio as presented in Figure 10 (*ϕ* = 5 mm ⇒
*ϕ* = 12 mm ⇒
*ϕ* = 25 mm).

## 4. Conclusions

The original intention of our research was in the field of aircraft tail antennas. It was based on the analysis of the influence of an aircraft antenna’s position on its directional properties. We expected standard outputs corresponding to the conventional knowledge of antenna theory, which is based on cylindrical antennas. By summarizing the measured results, we discovered some interesting properties of aircraft tail antennas modelled as flat microwave dipoles. This is due to the fact that flat microwave antennas behave differently in the region of resonant frequencies than generally expected.

We presented this difference by comparing the resonant properties of cylindrical and flat antennas. The outputs from the measurement of tail antennas as flat microwave dipoles showed that the graphs of their resonance dependence on length intersect at one point. This fact was subjected to a theoretical analysis on the generally known course of the dependence of the antenna impedance on its ratio *l/λ*. However, it was based only on the measured so-called antenna resonance points, i.e., when the reactance value of the antenna is equal to zero. It was the analysis of these points that showed that this is a new hitherto unknown view of a certain area of antenna theory applicable to modern, non-cylindrical antenna designs.

Further measurements (which are not part of this article due to space limitations) have shown that the position of the significant crossover point changes depending on the thickness of the flat dipole and is lost with a cylindrical shape of the dipole. In addition, it was found that a significant point is also formed on the microwave antennas at a second, parallel, half-wave resonance, and for dipoles with capacitive load. As follows from the presented facts, new possibilities have opened up in the design and use not only of aircraft tail antennas but also of other kinds of flat microwave antennas, such as those in automobiles and UAVs.

## Figures and Tables

**Figure 1 sensors-21-00939-f001:**
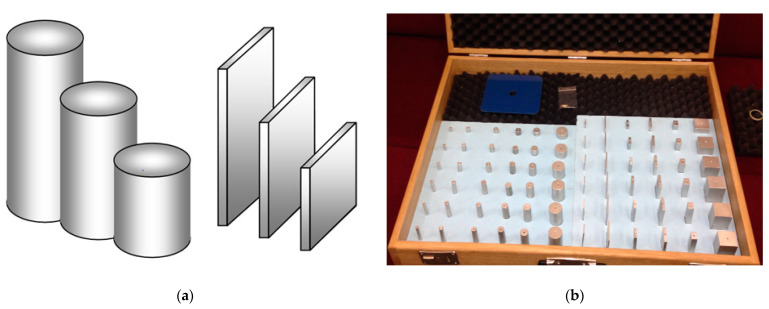
Measured antennas: (**a**) Illustration of cylindrical and flat antennas; (**b**) A dozen models produced of sets of antennas.

**Figure 2 sensors-21-00939-f002:**
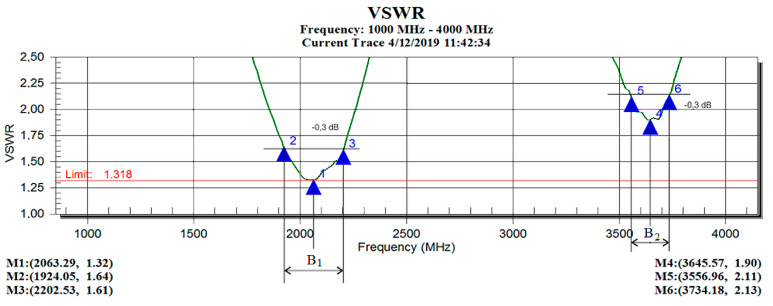
VSWR antenna recording for determining the value of the resonance frequency.

**Figure 3 sensors-21-00939-f003:**
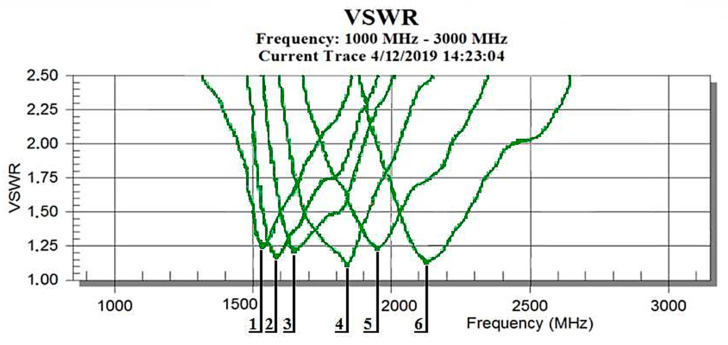
Example of one set of resonance frequencies of the antennas.

**Figure 4 sensors-21-00939-f004:**
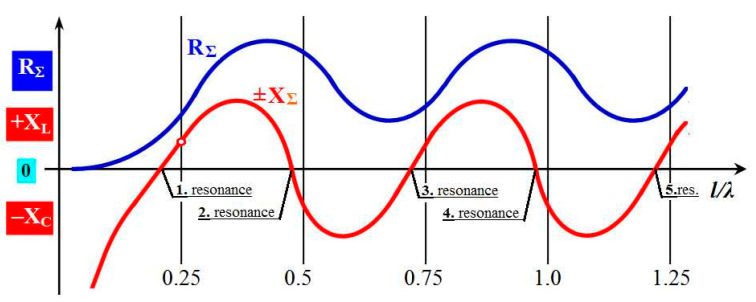
The simplified general course of antenna impedance.

**Figure 5 sensors-21-00939-f005:**
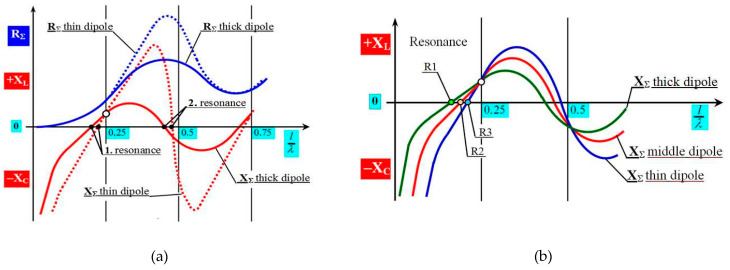
Influence of antenna conductor thickness on its impedance: (**a**) Antenna thickness and its impedance in the area of the 1st and 2nd resonance; (**b**) Antenna thickness and position of reactance resonance points.

**Figure 6 sensors-21-00939-f006:**
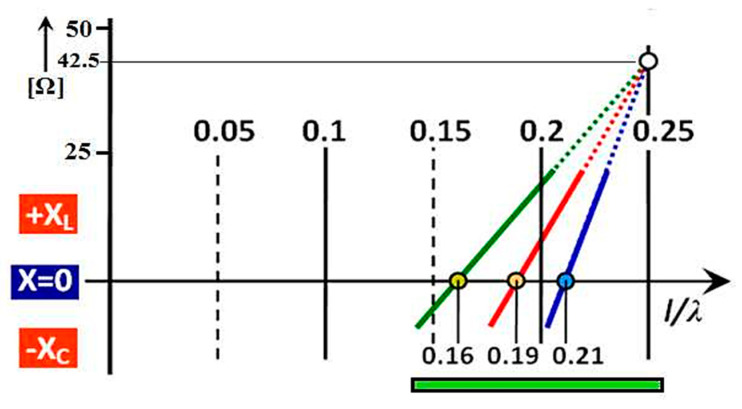
Determining the steepness of the reactance line and the position of the resonant points.

**Figure 7 sensors-21-00939-f007:**
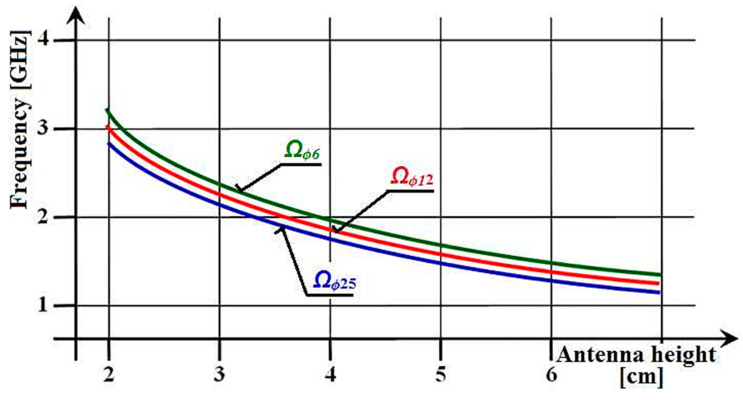
The graphical course of resonant frequencies depending on the slenderness of the dipole.

**Figure 8 sensors-21-00939-f008:**
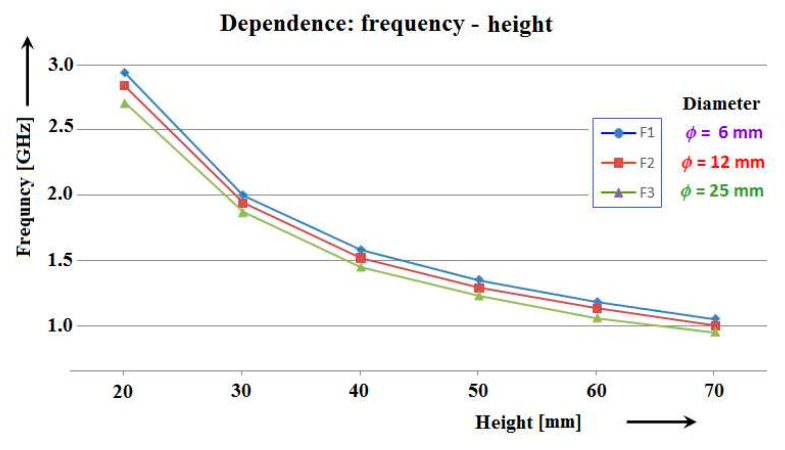
Dependence of resonant frequencies of cylindrical antennas on their length.

**Figure 9 sensors-21-00939-f009:**
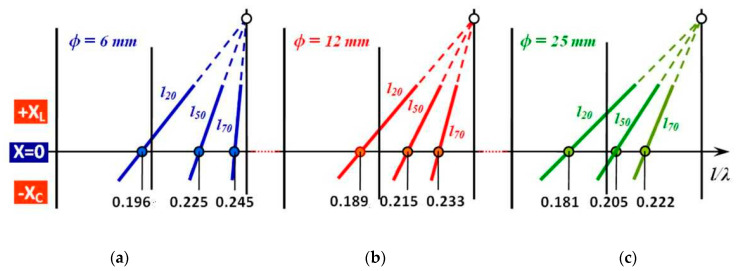
Change of position of resonance points $l/\lambda_{res}$ of cylindrical antennas when changing the slimness coefficient Ω: (**a**) Resonance points of very thin antennas *ϕ* = 5 mm; (**b**) Resonance points of medium-thick antennas *ϕ* = 10 mm; (**c**) resonance points of very thick antennas *ϕ* = 30 mm.

**Figure 10 sensors-21-00939-f010:**
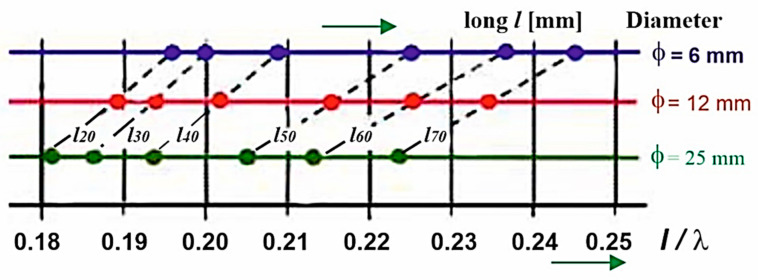
Concentrated distribution of the position of the resonant points of a set of cylindrical antennas.

**Figure 11 sensors-21-00939-f011:**
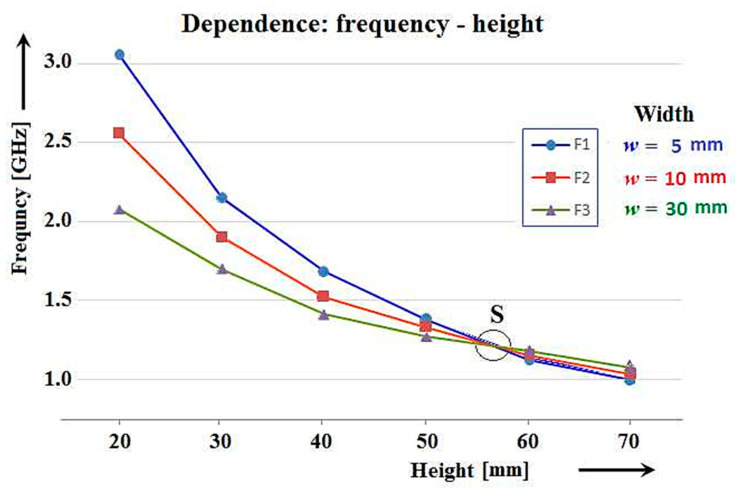
Dependence of resonant frequencies of flat microwave antennas on their length showing the significant crossover point *S*.

**Figure 12 sensors-21-00939-f012:**
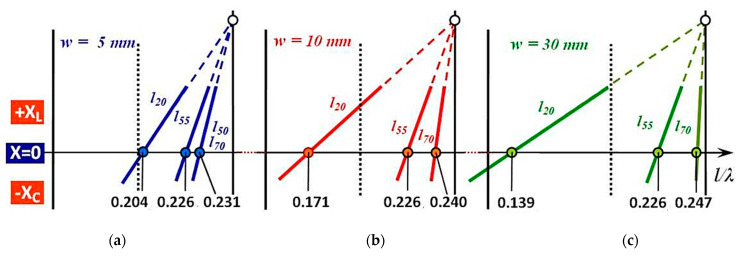
Change of position of resonance points $l/\lambda_{res}$ of flat microwave antennas when changing the slimness coefficient Ω**: (**a**) Resonance points of very narrow antennas w = 5 mm; (**b**) Resonance points of medium-wide antennas w = 10 mm; (**c**) Resonance points of very wide antennas w = 30 mm.

**Figure 13 sensors-21-00939-f013:**
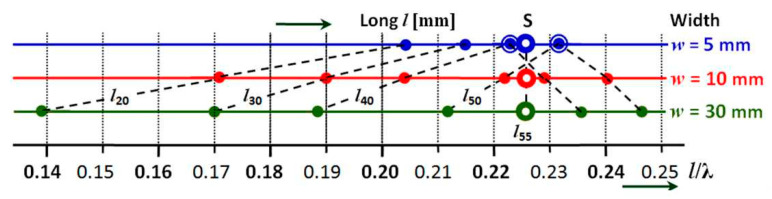
Concentrated distribution of the position of resonant points of a set of flat microwave antennas again showing the significant crossover point *S*.

**Table 1 sensors-21-00939-t001:** Values of resonance points $l/\lambda_{res}$ of cylindrical antennas.

20	30	40	50	60	70	*l* [mm]
0.196	0.200	0.209	0.225	0.237	0.245	(*ϕ* = 6 mm)
0.189	0.194	0.202	0.215	0.225	0.234	(*ϕ* = 12 mm)
0.181	0.187	0.193	0.205	0.213	0.223	(*ϕ* = 25 mm)

**Table 2 sensors-21-00939-t002:** Values of resonance points $l/\lambda_{res}$ of cylindrical antennas.

$\mathrm{The}\;\mathrm{Effect}\;\mathrm{of}\;\mathrm{Changing}\;\mathrm{the}\;\mathrm{Slimness}\;\mathrm{of}\;\mathrm{the}\;\mathrm{Antenna}\;(\mathrm{Changing}\;\mathrm{Diameter}\;\phi )\;\mathrm{at}\;\mathrm{the}\;\mathrm{Same}\;\mathrm{Height}\;l,\;\mathrm{on}\;\mathrm{the}\;\mathrm{Change}\;\mathrm{of}\;\mathrm{the}\;l/\lambda_{res}\mathrm{Ratio}$	$\mathrm{The}\;\mathrm{Effect}\;\mathrm{of}\;\mathrm{Changing}\;\mathrm{the}\;\mathrm{Slimness}\;\mathrm{of}\;\mathrm{the}\;\mathrm{Antenna}\;(\mathrm{Changing}\;\mathrm{Height}\;l)\;\mathrm{at}\;\mathrm{the}\;\mathrm{Same}\;\mathrm{Diameter}\;\phi ,\;\mathrm{on}\;\mathrm{the}\;\mathrm{Change}\;\mathrm{of}\;\mathrm{the}\;l/\lambda_{res}\mathrm{Ratio}$
Ω*_ϕ_*_6_*l*_20_ > Ω*_ϕ_*_12_ *l*_20_ > Ω*_ϕ_*_25_ *l*_20_	Ω*_ϕ_*_6_*l*_20_ < Ω*_ϕ_*_6_ *l*_50_ < Ω*_ϕ_*_6_ *l*_70_
0.196 0.189 0.181	0.1960 0.225 0.245
Ω*_ϕ_*_6_*l*_50_ > Ω*_ϕ_*_12_ *l*_50_ > Ω*_ϕ_*_25_ *l*_50_	Ω*_ϕ_*_12_*l*_20_ < Ω*_ϕ_*_12_ *l*_50_ < Ω*_ϕ_*_12_ *l*_70_
0.225 0.215 0.205	0.189 0.215 0.233
Ω*_ϕ_*_6_*l*_70_, > Ω*_ϕ_*_12_ *l*_70_ > Ω*_ϕ_*_25_ *l*_70_	Ω*_ϕ_*_25_*l*_20_ < Ω*_ϕ_*_25_ *l*_50_ < Ω*_ϕ_*_25_ *l*_70_
0.245 0.233 0.222	0.181 0.205 0.222

**Table 3 sensors-21-00939-t003:** Resonance point values $l/\lambda_{res}$ of flat microwave antennas.

20	30	40	50	55	60	70	*l* [mm]
0.204	0.215	0.224	0.231	0.226	0.224	0.231	*l*/λ (w = 5 mm)
0.171	0.190	0.203	0.222	0.226	0.229	0.240	*l*/λ (w = 10mm)
0.139	0.170	0.188	0.212	0.226	0.236	0.247	*l*/λ (w = 30 mm)

**Table 4 sensors-21-00939-t004:** Values of resonance points $l/\lambda_{res}$ of flat microwave antennas.

Influence of the change Ω** of the flat antenna on the change of the ratio *l**/λ_res_* due to changing *w* at the same *l*.	Influence of the change Ω** of the flat antenna on the change of the ratio *l**/λ_res_* due to changing *l* at the same w.
Ω*_w5_ l*_20_ > Ω*_w_*_10_ *l*_20_ > Ω*_w30_ l*_20_	Ω _w5_ l_20_ < Ω _w5_ l_55_ < Ω _w5_ l_70_
0.204 0.171 0.139	0.204 0.226 0.231
Ω*_w5_ l*_55_ = Ω*_w_*_10_ *l*_55_ = Ω*_w30_ l*_55_	Ω _w10_ l_20_ < Ω _w10_ l_55_ < Ω _w10_ l_70_
0.226 0.226 0.226	0.171 0.226 0.240
Ω*_w5_ l*_70_ < Ω*_w_*_10_ *l*_70_ < Ω*_w30_ l*_70_	Ω _w30_ l_20_ < Ω _w30_ l_55_ < Ω _w30_ l_70_
0.231 0.240 0.247	0.139 0.226 0.247

## Data Availability

Data sharing not applicable.

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
