# Peer review of "Specific Resonant Properties of Non-Symmetrical Microwave Antennas"

_sensors, 2021, doi:10.3390/s21030939_

Round 1

Reviewer 1 Report

The authors presented some air-born antennas initially and some specific properties of cylindrical antennas in the later section. This study is not well designed and executed. The ideas and data have not been reported concisely, and logically.  It is difficult to follow the manuscript. There is no link between different sections of the manuscript. The claim made in the last section about the new resonant properties of the cylindrical antennas which are stated as a breakthrough in antenna engineering lacks supported data and theory. Moreover, most of the references are not relevant. For example, see [1]-[3]. These references are different from the statements made in the manuscript.  In conclusion, the Reviewer does not recommend it for publication.

Author Response

Dear Reviewer,

Thank you very much for reviewing our paper, and for your comments and detailed suggestions helped us to improve our paper. We hope that we have understood them well. In accordance with your comments and suggestions as well as the suggestions of the other reviewers, we have improved significantly our manuscript as much as possible within the allotted time. All the changes/improvements are also highlighted in the Highlighted Changes file.

Comment of Reviewer 1:

The authors presented some air-born antennas initially and some specific properties of cylindrical antennas in the later section. This study is not well designed and executed. The ideas and data have not been reported concisely, and logically.  It is difficult to follow the manuscript. There is no link between different sections of the manuscript. The claim made in the last section about the new resonant properties of the cylindrical antennas which are stated as a breakthrough in antenna engineering lacks supported data and theory. Moreover, most of the references are not relevant. For example, see [1]-[3]. These references are different from the statements made in the manuscript.  In conclusion, the Reviewer does not recommend it for publication.

Response to Reviewer 1:

We have reorganized and rewritten the manuscript.

In the first four short sentences of the text, the reviewer expresses his first reservation on the inappropriate structure of the article. When writing the article, the authors considered it important to explain to the reader the motivation of the performed measurement, during which new resonant properties of flat microwave antennas were accidentally revealed. The authors devoted this introductory to two short sections (2nd and 3rd) to this explanation. It turned out that this was not a suitable solution, because it distracted the reader from the essence of the problem, which was explained only in the final section. From this point of view, the authors agree that this was an inappropriate structure of the article and that an amendment to the article was needed.

The original, introductory two sections (2nd and 3rd) have been deleted and their problems are mentioned in a very shortened version in the introduction. This shortened, simplified and clarified the article in order to help readers understand the idea of the article.

The reviewer's reservation is: „The claim made in the last section about the new resonant properties of the cylindrical antennas which are stated as a breakthrough in antenna engineering lacks supported data and theory.“

Yes, we agree that “breakthrough” is perhaps too strong a claim in such a large field, so we have softened the wording used in the Abstract, Introduction and Conclusion to more modestly reflect the paper’s contribution. Nonetheless, as researchers in the area we found the experimental results interesting and potentially useful to practitioners in antenna design, and therefore well worthy of dissemination. Two-thirds of the article is devoted to the explanation of antenna theory, the presentation of experimental evidence and their analysis. Since this is an explanation of something new in the field of antenna theory, the authors want to share this knowledge with the professional public through an article.

There are three short sentences at the end of the text by which the reviewer expresses his third reservation about an inappropriate list of cross references to the literature. After reviewing the list of links, the authors agree to this reservation. The creation of a not very suitable list of links was caused by two factors. The first is the time pressure for timely submission of the article. The authors present newly discovered knowledge in the field of antenna theory, which is still very little published and there are few sources of information. Therefore, there is literature in the list of references that only approaches the topic. In each case, the authors revised the list of references in the form of reductions and additions at the reviewer's initiative.

Links [1], [2], [3], [9],  have been removed and replaced by new citations [1], [2], [3], [9] and have been added new links [14],  [15], [16], [17], [18], [19], [20], [21],.

We would like to thank the reviewer for critical comments on the article, based on which we significantly re-evaluated the content structure of the article and its professional content.

Reviewer 2 Report

COMMENTS:

Many points in the manuscript must be clarified before its publication.

1) Lines 22 and 50: The first line states that “The analysis of the resonant properties of non-symmetric antennas was performed in the band from 1 GHz to 3 GHz”, whereas the second line states that “Analysis of the resonance characteristics of non-symmetrical microstrip antennas was performed in frequency ranges from 1 GHz to 4 GHz”. Which piece of information is correct?

2) Lines 87 and 88: Reference [8] does not bring the radiation pattern presented in Figure 1(b), as said in the text.

3) The authors should check the use of the term “damping chamber” along the text.

4) Line 105: Please check the symbol for the aircraft model’s size.

5) The authors state that the antennas they studied in the paper are “microstrip antennas”. Which was the microwave laminate used to build those “microstrip antennas”? Please check in the antenna literature the composition of conventional microstrip antennas.

6) Line 50: “During the measurements, the VSWR value of the individual antennas was evaluated in the frequency range from 1 GHz to 3 GHz”. However, Figure 4 shows measurements from 1 GHz to 4 GHz.

7) Please check equation (1).

8) Line 178: It should be better explained why “at frequencies in the UHF band, the losses are negligible”.

9) Lines 153, 154, and 196: In the first two lines, resonance occurs when the VSWR assumes a local minimum. On the other hand, in the last line, resonance occurs when the imaginary part of the antenna input impedance is equal to zero. This point should be clarified. Are they equivalent criteria?

10) The decimal separator used in the graphs of Figures 6 and 7 is comma instead of point.

11) How the curves in Figure 7 were derived? The authors should include either a reference or an explanation on how they were evaluated (equations, numerical methods, etc.).

12) Figure 13 label is written as Figure 1.

13) Line 399: “bandwidth w = 5 mm”. It is not common a bandwidth expressed in mm.

14) Could the results obtained from measurements be derived from numerical simulations using commercial software, for example? Were the antennas installed on an aircraft model during measurements? If so, in which part of the model they were installed?

15) A theoretical analysis should be presented in the paper to justify the experimental observations, since there are numeral techniques, for example, suitable to the analysis of the antennas studied in the manuscript.

Author Response

We would like to thank the reviewer for critical comments on the article, based on which we significantly re-evaluated the content structure of the article and its professional content.

Reviewer 3 Report

in this paper, authors have been analyzed the effect of aircraft antenna’s position on its directional properties. This study demonstrates that microstrip dipoles have a resonant frequency different from that expected according to the known theories of classical cylindrical antennas.

The analysis is based on measured data using a Scaled model of Embraer Phenom aircraft. This could helps Engineers to perform and design Airborne Antennas.

The novelty is clearly demonstrated and the contribution is a good candidate for aviation applications:

However, the article is long and not easy to read and follow by readers. The following issues must be addressed:

  • Propose a new simple paper organization with the aim to help readers understanding the paper idea.
  • Add a simulation of one or two study scenarios (Figure 10 to Figure 15) to validate that resonant frequencies versus  length crossover at one point for microstrip antenna models.
  • The first sentence in Line 134 is not very clear from my understanding, 'If the antenna model is to perform ...'
  • Equation 3 Line 234-235 needs to be referenced from literature, please add at least one reference.
  • Authors should add more recent references.

Author Response

Dear Reviewer,

Thank you very much for reviewing our paper, and for your comments and suggestions. We hope that we have understood them well. In accordance with your comments and suggestions as well as suggestions of other reviewers, we have improved significantly our manuscript as much as possible within the allotted time. All the changes/improvements are also highlighted in the Highlighted Changes file. We ask you to consider the manuscript as a conceptual paper.

Reviewer 3:

In this paper, authors have been analyzed the effect of aircraft antenna’s position on its directional properties. This study demonstrates that microstrip dipoles have a resonant frequency different from that expected according to the known theories of classical cylindrical antennas.

The analysis is based on measured data using a Scaled model of Embraer Phenom aircraft. This could help Engineers to perform and design Airborne Antennas.

The novelty is clearly demonstrated and the contribution is a good candidate for aviation applications:

However, the article is long and not easy to read and follow by readers. The following issues must be addressed:

Propose a new simple paper organization to help readers understanding the paper idea.

Response to Reviewer 3:

We have reorganized and rewritten the manuscript.

When writing the article, the authors considered it important to explain the motivation of the performed measurement, during which new resonant properties of microwave flat antennas were revealed. Therefore, the first two short chapters were devoted to this problem. It turned out that this was not a good solution because it distracted the reader from the essence of the problem.

The original, first two chapters (2nd and 3rd) have been deleted. This shortened, simplified and clarified the article to help readers understand the idea of the article.

Reviewer 3:

Add a simulation of one or two study scenarios (Figure 10 to Figure 15) to validate that resonant frequency versus length crossover at one point for microstrip antenna models.

Response to Reviewer 3:

Given the very short time available to the authors and the epidemiological situation of COVID 19 in our country (after the declaration of lockdown), this is not currently possible. Thank you for understanding. The authors continue to address this issue and want to come up with an article later that will use numerical techniques. The numerical way of analyzing a given problem assumes that this could lead to concrete proposals for the use of this phenomenon in practice.

The authors state that this is a newly discovered problem, so it is quite likely that current programs will not respond in this way. So far, we have tried unsuccessfully to solve it in the FEKO program.

Reviewer 3:

The first sentence in Line 134 is not very clear from my understanding, 'If the antenna model is to perform ...'

Response to Reviewer 3:

This sentence has been modified as follows:

"If the antenna model is to radiate around the model aircraft in the same way as the corresponding antenna on an actual aircraft, both antennas must have approximately the same shape, proportional dimensions, similar electrical properties, etc."

It has been inserted into the modified article in the text in (Line 74).

Reviewer 3:

Equation 3 Line 234-235 needs to be referenced from literature, please add at least one reference.

Response to Reviewer 3:

The theory presented in the article is presented, for example, on the pages of a Wikipedia on the topic „Dipole antenna“ https://en.wikipedia.org/wiki/Dipole_antenna in Chapter „Dipole characteristics“ in point „Impedance of dipoles of various lengths“. The figure shows the impedance value of the half-wave dipole shown in the text ZA = 73 + j43 [Ω]. The reactance value is +43 Ω, Reactance this point passes through the course of each antenna element. In the literature [20] „Stutzman, Warren L.; Thiele, Gary A. (2012). Antenna Theory and Design. John Wiley and Sons. pp. 156. ISBN 978-0470576649“ page 156 shows the exact impedance value ZA = 73,1 + j42,3 [Ω].

So, Reference [20] has been added here in the current modified article (Line 168).

Reviewer 3:

Authors should add more recent references.

Response to Reviewer 3:

We have added the following recent references [6],[[14],[[15],[[16],[17],[18], [19],[[20],[and [21].

We would like to thank the reviewer for critical comments on the article, based on which we significantly re-evaluated the content structure of the article and its professional content.

Round 2

Reviewer 1 Report

I went through the response to the Reviewer’s comments and revised manuscript. The authors have addressed all the concerns. The manuscript can be accepted for publication.

Reviewer 3 Report

The paper has been improved and can be accepted for publication.